# Effect of Size on Hydrogen Adsorption on the Surface of Deposited Gold Nanoparticles

**DOI:** 10.3390/nano9030344

**Published:** 2019-03-03

**Authors:** Andrey Gatin, Maxim Grishin, Nadezhda Dokhlikova, Sergey Ozerin, Sergey Sarvadii, Vasiliy Kharitonov, Boris Shub

**Affiliations:** Semenov Institute of Chemical Physics Russian Academy of Sciences, 4 Kosygin street, Moscow 119991, Russia; akgatin@yandex.ru (A.G.); dohlikovanv@gmail.com (N.D.); sergeoz@yandex.ru (S.O.); sarvadiy15@mail.ru (S.S.); vch.ost@mail.ru (V.K.); bshub@mail.ru (B.S.)

**Keywords:** gold nanoparticles, size effects, hydrogen adsorption, scanning tunneling microscopy and spectroscopy

## Abstract

An experimental study of molecular hydrogen adsorption on single gold nanoparticles of various sizes deposited on the surface of highly oriented pyrolytic graphite (HOPG) was carried out by means of scanning tunneling microscopy and spectroscopy. The effect of size on the HOPG/Au system was established. Hydrogen was dissociatively chemisorbed on the surface of gold nanoparticles with an average size of 5–6 nanometers. An increase in the size of nanoparticles to 10 nm or more led to hydrogen chemisorption being inhibited and unable to be detected.

## 1. Introduction

Gold is a perfect example of a material that is inert in its massive form and highly active in the nanoscale form. In particular, gold nanoparticles are used to create catalysts for low-temperature oxidation of CO [1], for selective oxidative coupling of methanol to methyl formate [2], and for hydroxylamine oxidation [3]. Gold nanoparticles are also used as catalysts for selective isomerization of epoxides to allylic alcohols [4], for benzylation of aromatics [5], for the production of vinyl chloride and vinyl acetate monomers [6], and in some other substances. Gold nanoparticles are used in catalytic reactions of alkene hydrogenation [7], in chemoselective hydrogenation of crotonaldehyde to crotyl alcohol [8], in chemoselective hydrogenation of nitro compounds [9], and in heterogeneous hydroformylation of olefins [10]. Gold-based nanocatalysts are also used for reduction of nitro compounds [11,12].

Catalysts based on Au nanoparticles have been proven to be effective in the reactions of hydrogenation with hydrogen transfer, such as reduction of carbonyl compounds [13] and amines [14] and for hydrochlorination of alkynes [15]. Dissociative adsorption of molecular hydrogen and interaction of the generated atomic hydrogen with the surface of nanoparticles play an important role in these processes. With the use of calorimetry, transmission electron microscopy (TEM), and X-ray diffraction (XRD), direct synthesis of water from O_2_ and H_2_ was found to occur over silica-supported gold nanoparticles at 383–433 K and a pressure of the order of several kilopascals [16]. The results of the corresponding density functional theory (DFT) calculations have also been presented. The results obtained for Al_2_O_3_-supported gold nanoparticles [17,18] confirmed the conclusion of a previous report [16] about the interaction of gold and hydrogen. It has been emphasized that Au atoms with low coordination numbers and/or defects such as vacancies and impurity atoms can be sites of hydrogen adsorption and dissociation. However, as shown by X-ray photoelectron spectroscopy (XPS) and mass spectroscopy in a previous report [19], hydrogen dissociation takes place on the perimeter interfaces between gold and TiO_2_ for a catalyst based on gold particles on a rutile TiO_2_ support. At the same time, the calculation results [20] differ from the assumption mentioned above. It has specifically been stated that the sites of H_2_ dissociation must be low-coordinated gold atoms not directly bound to the stoichiometric and reduced TiO_2_ support. In a previous report [21], hydrogen atoms were suggested to be generated as a result of H_2_ dissociation on low-coordinated gold atoms with their further migration to the substrate.

Experimentally, chemisorption of hydrogen on the surface of gold nanoparticles was discovered relatively recently [22,23,24,25,26]. In our previous works [25,26], using scanning tunneling microscopy and spectroscopy, we observed dissociative adsorption of hydrogen on gold nanoparticles deposited on graphite. The reasons for the increase in activity with the reduction in particle size to nanometers may be various, including size quantization, nanoparticles being charged due to interaction with the substrate, an increase in the specific fraction of low-coordinated gold atoms, etc. This study shows the effect of size on the ability of nanoparticles to dissociatively adsorb hydrogen on their surface.

## 2. Materials and Methods

The experiments were carried out at a facility consisting of a scanning tunneling microscope (STM, Omicron NanoTechnology, Taunusstein, Germany), Auger spectrometer (Omicron NanoTechnology), quadrupole mass spectrometer (Hiden Analytical Limited, Warrington, UK), and auxiliary equipment at 300 K and a residual gas pressure of 2 × 10^−10^ Torr. As STM probes, we used tips made of platinum–iridium and tungsten wires using the standard methods.

Nanoparticles were deposited on the surface of HOPG using the impregnation method. To achieve this, an aqueous solution of HAuCl_4_ with a metal concentration of 5 × 10^−6^ g/mL was applied over the surface of the substrate. Then, the sample was dried, placed in a vacuum chamber, and annealed in ultra-high vacuum at 500–750 K for several hours. The required duration of annealing was established using the results of an STM study of the surface morphology of the sample.

The morphology and electronic structure of the surface of the sample at the level of single nanoparticles and the results of their modification due to interaction with the adsorbate were both determined using topographic and spectroscopic STM measurements. It is known that the nanocontact formed by a metal sample and conducting tip demonstrates an S-shaped dependence of the STM tunneling current on voltage (volt–ampere characteristic, VAC) [27]. A change in the elemental composition of a sample, for instance, due to chemical reactions, can lead to the transition of its electronic structure from the metal to semiconductor type or to a significant decrease in the density of states in the vicinity of the Fermi level. This alteration results in the appearance of a zero current region within the S-shaped VAC curve attributed to the band gap of the material [28,29,30,31]. Thus, changes in the shape of the VAC curve can indicate a change in chemical composition over the surface of nanoparticles. Hereafter, the term “VAC of nanoparticles” refers to the VAC of the STM tunneling contact formed by the STM tip and the surface of gold nanoparticles placed on the graphite support, while the term “VAC of graphite” refers to the VAC of the tunneling contact of the STM tip with the pure graphite surface.

The analysis of the elemental composition of the samples surface was carried out by Auger spectroscopy. The obtained data were compared with the results of the spectroscopic STM measurements.

The gas composition at all stages of the experiment was controlled by the results of mass spectrometric measurements. In the experiments described below, the molecular hydrogen pressure did not exceed P = 1 × 10^−6^ Torr. The exposure value was measured in Langmuir, 1 L = 1 × 10^−6^ Torr × s.

## 3. Results and Discussion

In accordance with the above method, two samples were synthesized. The difference between them was in the duration of annealing in vacuum. The first sample was heated for 10 h, while the second one was heated for 30 h. As a result, a coating of gold nanoparticles was formed on the surface of graphite. The gold nanoparticles had a rounded shape. The distribution of the lateral nanoparticle size for the first sample had a maximum in the range of 4–6 nm, and the average nanoparticle height was about 1.5–2 nm. For the second sample, the distribution showed a maximum in the range of 10–15 nm, and the average nanoparticle height was about 3–6 nm. Examples of topography images of single nanoparticles are shown in Figure 1. Figure 1a shows an image of two single gold nanoparticles with the lateral size of about 5 nm, located at the edge of a graphene sheet. Figure 1b shows an image of a single nanoparticle with a diameter of about 10 nm, located in the defect-free area of the graphite surface.

Figure 2 shows topography (Figure 2a) and current (Figure 2b) images of a HOPG surface area with a gold nanoparticle deposited on it. The current image is a map of the values of the tunneling currents at a certain voltage value. Darker points correspond to larger values of the tunneling current. The current image gives an idea of the local electron density of the surface of the sample in an explicit form. Earlier [25], we had shown that the conductivity of pure gold nanoparticles, i.e., those not containing impurities and not covered with a layer of adsorbate, is slightly higher than the conductivity of the HOPG surface on which they are deposited. This statement is also true for the samples synthesized in the present work. Figure 2d shows the current–voltage dependencies of the tunneling current averaged over the surface of the particles (curve A) and over the HOPG surface (curve B). The regions of averaging are marked with rectangles in Figure 2c. One can see that VAC of nanoparticles exceeded VAC of HOPG in the absolute values of the tunneling current. The gold nanoparticles had no impurities and adsorbates on their surface, which was proven by both the Auger spectra containing peaks corresponding only to gold and carbon and by the smooth S-shape of VAC typical for a pure metal–metal nanocontact (see Figure 2d).

Figure 3 shows topography (Figure 3a) and current (Figure 3b) images of the HOPG surface area containing a gold nanoparticle of 12 nm in size. One can see that particles of different sizes correspond to darker areas in the current image; thus, the conductivity of nanoparticles exceeded the conductivity of HOPG regardless of the nanoparticle size.

The purpose of this study was to establish a correlation between the nanoparticle size and their adsorption properties in relation to hydrogen. Both samples were exposed to H_2_ (2000 L exposure). In a previous report [25], we had demonstrated that hydrogen was adsorbed dissociatively on the surface of nanoparticles of 5–6 nm in size. In the case of such nanoparticles, hydrogen adsorption leads to a drastic change in the electronic structure of the surface of nanoparticles and significantly decreases the conductivity of the STM tunneling contact. In a previous report [32], deuterium adsorption on the surface of gold nanoparticles with a size of 5–6 nm was experimentally and theoretically investigated in detail. It was shown that, at low exposures, the interface of gold nanoparticles and HOPG support was the preferred place for deuterium adsorption. With the further increase in exposure and saturation of the interface adsorption sites, deuterium gradually covered the rest of the gold nanoparticle surface from periphery to center. Complete coverage was achieved at the exposure of 1800 L. Therefore, the exposure of 2000 L would be enough to completely cover the nanoparticle surface with adsorbed hydrogen.

Figure 4 shows topography (Figure 4a,c) and current (Figure 4b,d) images of two surface areas of HOPG with gold nanoparticles of different sizes after their exposure to hydrogen. From the analysis of the current image (Figure 4b), one can see that, after hydrogen adsorption, the conductivity of a 5 nm nanoparticle became lower in its absolute value than the conductivity of HOPG; the area of the current image corresponding to the nanoparticle became lighter. In a previous report [33], adsorption of atomic hydrogen on the surface of unsupported gold nanoclusters was shown to result in a decrease in the density of states in the vicinity of the Fermi level, which entails a decrease in the conductivity of a gold nanocluster. Therefore, for the case of nanoparticles of 5 nm in size, we can conclude that a layer of chemisorbed atomic hydrogen was formed on the surface of the nanoparticles, which in turn led to a decrease in the conductivity of the tunneling nanocontact containing gold nanoparticles. We use the word “layer” to denote the quantity of adsorbed hydrogen, which provides the effect of visible changes in conductivity, i.e., in local electron density, over the entire surface of the nanoparticle. This quantity of hydrogen is actually different from the one necessary for the monolayer formation.

In the case of nanoparticles with a size of 12 nm (Figure 4c), the exposure of the sample to hydrogen did not lead to any noticeable changes in the electronic structure of nanoparticles. As with the exposure to hydrogen, the conductivity of nanoparticles remained higher than the conductivity of the HOPG support (Figure 4d), i.e., hydrogen chemisorption was not observed over the entire surface of the 10 nm gold nanoparticle unlike in the case of the 5 nm nanoparticle. Nevertheless, we really cannot exclude the adsorption of a countable number of hydrogen atoms on the surface of large nanoparticles, for example, on any defects. However, this quantity will be significantly less in comparison to the case of small nanoparticles. Having processed the data obtained, we can claim that hydrogen chemisorption is not observed on the surface of gold nanoparticles with a size of more than 10 nm.

In a previous report [32], it was shown that the existence of a stable form of atomic hydrogen adsorption on the surface of gold nanoparticles is associated with an increase in the contribution of surface states to the formation of Au–H bond. An increase in the size of nanoparticles leads to a decrease in the influence of surface states on the formation of Au–H bond. Apparently, a nanoparticle size of about 10 nm is a threshold value, above which hydrogen chemisorption on the surface of gold nanoparticles becomes energetically disadvantageous.

Thus, we can conclude that surface states and their influence on the binding energy of hydrogen with nanostructured gold provide stable forms of adsorption observed in our experiments for nanoparticles with a size of 5–6 nm. At the same time, adsorption of atomic hydrogen cannot be detected for nanoparticles with a size of more than 10 nm.

## 4. Conclusions

Molecular hydrogen adsorption was studied experimentally for gold nanoparticles of different sizes deposited on the surface of a HOPG support. The effect of size was established for the Au–HOPG nanostructured system. Hydrogen was dissociatively chemisorbed on the surface of gold nanoparticles with a size of 5–6 nm, while hydrogen chemisorption was not observed for gold nanoparticles with a size of ≥10 nm.

## Figures and Tables

**Figure 1 nanomaterials-09-00344-f001:**
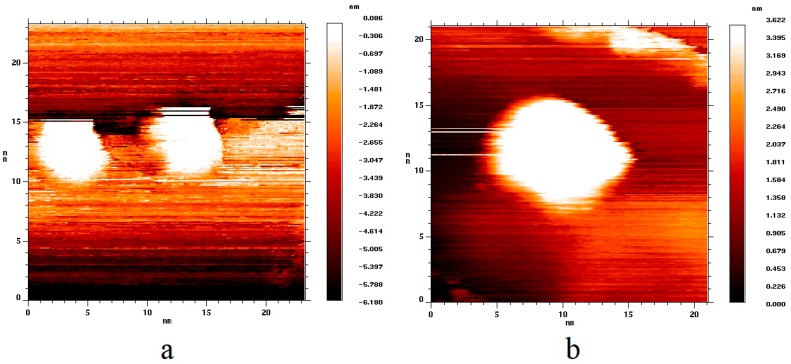
Topography images of the highly oriented pyrolytic graphite (HOPG) surface with deposited gold nanoparticles. (**a**) Sample with 5 nm nanoparticles (U_bias_ = 2.0 V, I_tunnel_ = 2.9 nA); (**b**) sample with 10 nm nanoparticles (U_bias_ = 2.1 V, I_tunnel_ = 3.1 nA).

**Figure 2 nanomaterials-09-00344-f002:**
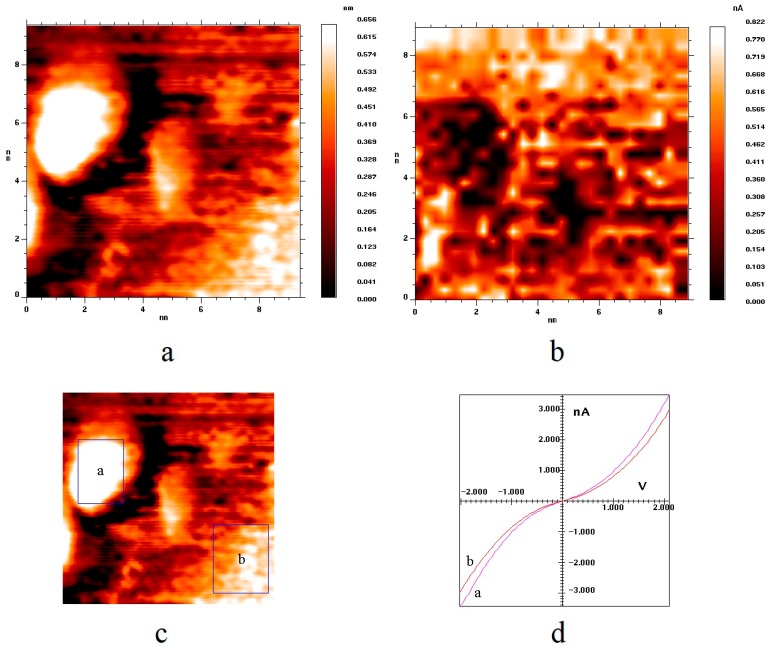
(**a**,**c**) Topography images of HOPG surface area containing a gold nanoparticle; (**b**) current image of the same surface area; (**d**) volt–ampere characteristic (VAC) of the tunneling currents, averaged over the surface areas indicated by the rectangles in Figure 2c. U_bias_ = 2.1 V, I_tunnel_ = 2.9 nA.

**Figure 3 nanomaterials-09-00344-f003:**
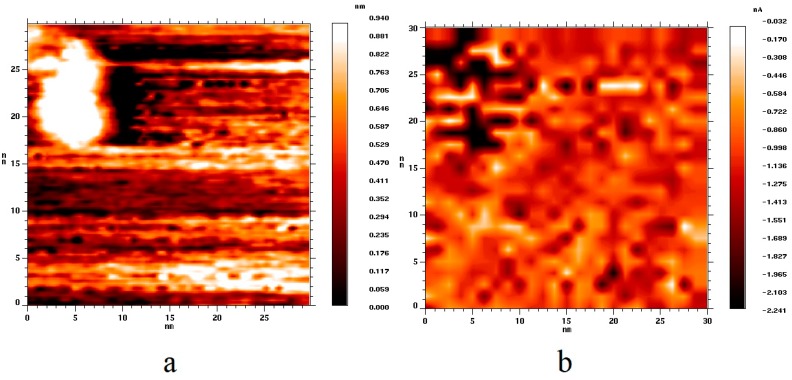
(**a**) Topography image of the HOPG surface area with a deposited gold nanoparticle; (**b**) current image of the same surface area. U_bias_ = 1.9 V, I_tunnel_ = 3.1 nA.

**Figure 4 nanomaterials-09-00344-f004:**
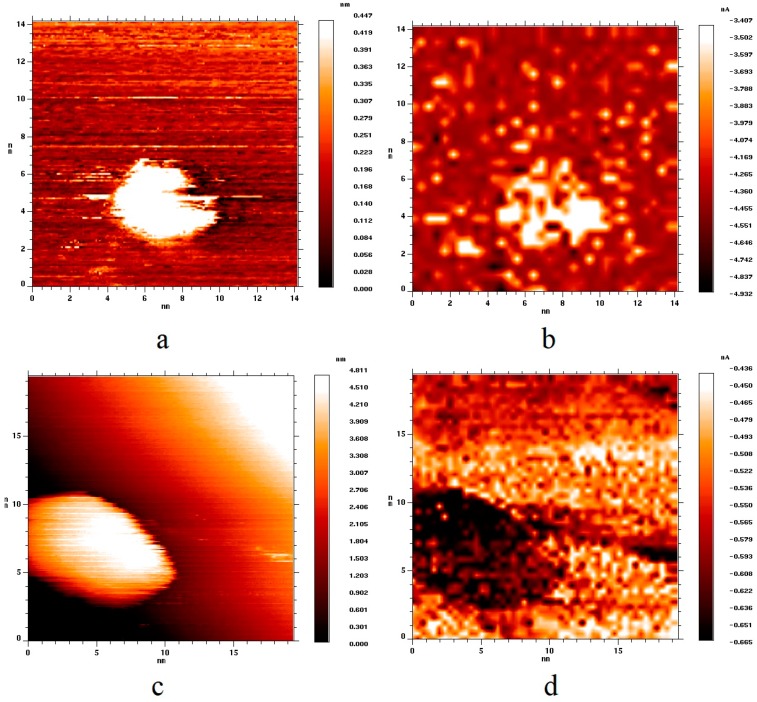
(**a**,**c**) Topography images of two HOPG surface areas with deposited gold nanoparticles of various sizes after their exposure to molecular hydrogen; (**b**,**d**) current images of the same surface areas. (**a**,**b**) U_bias_ = 2.0 V, I_tunnel_ = 3.1 nA; (**c**,**d**) U_bias_ = 2.1 V, I_tunnel_ = 3.1 nA.

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
