# Peer review of "Effect of Size on Hydrogen Adsorption on the Surface of Deposited Gold Nanoparticles"

_nanomaterials, 2019, doi:10.3390/nano9030344_

Round 1
Reviewer 1 Report
This article deals with an STM investigation of the adsorption of hydrogen on supported gold nanoparticles (GNPs) of different sizes grown on HOPG.
In complement to a couple of articles concerning this issue, already published by the same authors, the present paper, based on the comparison between several GNPs, shows a difference as a function of the size of the GNPs, indicating a dissociative adsorption for a GNP of 5 nm, and no adsorption for a NP of 12 nm.
The authors compare the current images obtained on these two GNPs, without or with H2 exposure. In the former case, there is an increase of current, evidencing the metallic character of the GNPs. In the latter case, after exposure to 200 L of H2, the same tendnacy is observed for the larger GNP of 12 nm, while a decrease of current is obtained for the 4 nm GNP. This observed difference demonstrates different interaction of H2 with the larger and the smaller GNP.
Although these results deserve to be published in Nanomaterials, the manuscript is not well written, nor the discussion well conducted, and requires significant changes before it can be considered for publication.
The main criticism is the discussion drawn between lines 140 and 155:
1. The observation of change in the conductivity of the small (4 nm) GNP clearly indicates some hydrogen adsorption on the GNP, but this observation cannot leads to the conclusion that a “layer of chemisorbed atomic hydrogen is formed”.
2. The fact that there is no apparent effect of exposure to H2 on the conductivity of the 12 nm GNP does not mean that there is no hydrogen chemisorption. It means that the possible H chemisorption does not change strongly the conductivity. It is quite possible that some H atoms are adsorbed, e.g. in low coordination Au atoms,, without a change of the electronic properties of the 12 nm GNP, large enough to be detectable by the current imaging. On the contrary, the same amount of adsorbed H on the smaller GNP (4 nm), which is the limit of a clear metallic behavior, can more strongly modify the current image, as shown experimentally.
3. In the present article, there are strong arguments related to ref. 36, and it is written (line 126): “so the exposure of 2000L is enough for completed covering of nanoparticles surface with adsorbed hydrogen”. This is a strong conclusion related to above point 1. Ref. 36 is also discussed and is used for concluding that 10 nm is a threshold value (lines 156-161). However, Ref. 36 is still not published, and therefore it is not possible to rely on its conclusion. The authors should, either wait for publication of ref. 36, or send a copy of this paper, if it is actually accepted, to the Referees.
There are also some less important criticisms, which have also to be answered.
4. Although the images of the NPs after H2 expoure dispaly a good resolution (fig.4), the resoultion is much poorer in fig. 2 and 3. It shoud be necessary to show topogrtaphic and current images with the same resolutions than in Fig.4 for small and large GNP, without H2 exposure.
5. Tunneling conditions for each image have to be given (STM bias, STM current).
6. The reference have to be checked : ref. 25, 27, 34 and 35 are the same : ref. 26 and 28 are the same.
7. Finally, the English writing is not very good and it should be improved.
Author Response
Many thanks for your valuable comments that helped us to improve the quality of the manuscript.
Point 1. The observation of change in the conductivity of the small (4 nm) GNP clearly indicates some hydrogen adsorption on the GNP, but this observation cannot leads to the conclusion that a “layer of chemisorbed atomic hydrogen is formed”.
Response 1. In our previous article [Dokhlikova N.V., Kolchenko N.N., Grishin M.V., Shub B.R., Electron delocalization in heterogeneous AunHm systems // Nanotechnologies in Russia, 11 (2016) 7], it was shown that the adsorption of hydrogen on the surface of a gold cluster significantly changes the electron density of states, projected onto gold atoms located close to the adsorbed hydrogen atom. The density of states, projected onto other atoms of the gold cluster, changes slightly. That is, the adsorption of hydrogen has the property of locality. It is the local density of the electron states of the sample that determines the conductivity of the tunneling nanocontact. Thus, the adsorption of hydrogen leads to a local change in the conductivity of the tunneling nanocontact. We use the word "layer" to denote the quantity of adsorbed hydrogen which provides visible effect – changes in conductivity, i.e. in local electron density – over the entire surface of the nanoparticle. This quantity of hydrogen is actually less than one necessary for monolayer formation. Relevant clarifications have been added to the manuscript (lines 154-157 in revised manuscript).
Point 2. The fact that there is no apparent effect of exposure to H2 on the conductivity of the 12 nm GNP does not mean that there is no hydrogen chemisorption. It means that the possible H chemisorption does not change strongly the conductivity. It is quite possible that some H atoms are adsorbed, e.g. in low coordination Au atoms,, without a change of the electronic properties of the 12 nm GNP, large enough to be detectable by the current imaging. On the contrary, the same amount of adsorbed H on the smaller GNP (4 nm), which is the limit of a clear metallic behavior, can more strongly modify the current image, as shown experimentally.
Response 2. According to the resonant chemisorption model [B. Hammer and J.K. Norskov, Why gold is the noblest of all the metals // Nature, 376 (1995) 238], massive gold is the most inert of all the metals due to the fact that the antibonding state – related to the interaction of the adatom level with the low-located d-zone – is located below the Fermi level and is completely filled. That is, a chemical bond is not formed. However, in the case of gold nanoparticles, the situation changes. In this case the well-known "size effect" manifests itself in such a way, that contribution of gold surface states to the whole binding energy Eads increases with the size reduction of nanoparticle. In the case of interaction between gold nanoparticle surface and adatom, for the binding energy between Eads we have :
Eads = (ρ(εd)Ed + ρ(εss)Ess)/(ρ(εd) + ρ(εss))
where ρ(εd) and ρ(εss) are the peak amplitudes of density of d-states and surface states, and Ed and Ess are corresponding interaction energies. The ρ(εd)/ρ(εss) ratio is negligible for the surface of bulk gold monocrystal, but increases rapidly with nanoparticle size reduction as does the fraction of low-coordinated atoms. Also note that Ess > Ed. First of all it is due to the location of surface states band. For noble metal, the peak ρ(εss) is ~0.5 eV lower than the Fermi level. In terms of two-states formalism it means that the antibonding state of Au-H "surface molecule" is unoccupied and does not compensate the bonding state contribution to the binding energy unlike it could happen in case of interaction with just the d-zone states. At low bias voltages the decisive contribution to binding energy is made by the interaction of H-adatom state with surface states of gold nanoparticle. Note that peak of surface states decreases as hydrogen atoms fill the nanoparticle surface. So it is due to the reduction of density of states in the vicinity of the Fermi level, that we can detect the acts of hydrogen dissociative adsorption on gold nanoparticles in STM-STS experiments. An increase in the size of nanoparticles leads to a decrease in the ratio of surface states and can indeed slightly smooth out the experimentally observed effect of conductivity decrease during hydrogen adsorption. However, if nanoparticles are enlarged to a size at which this effect becomes unnoticeable, hydrogen adsorption becomes generally impossible. Nevertheless, we really cannot exclude the adsorption of a countable number of hydrogen atoms on the surface of large nanoparticles, e.g. on any defects. But this quantity will be significantly less in comparison to the case of small nanoparticles. Appropriate clarification has been added to the manuscript (lines 167-169 in revised manuscript).
The theoretical aspects presented above are beyond the scope of the manuscript, and will be considered in detail in another article.
Point 3. In the present article, there are strong arguments related to ref. 36, and it is written (line 126): “so the exposure of 2000L is enough for completed covering of nanoparticles surface with adsorbed hydrogen”. This is a strong conclusion related to above point 1. Ref. 36 is also discussed and is used for concluding that 10 nm is a threshold value (lines 156-161). However, Ref. 36 is still not published, and therefore it is not possible to rely on its conclusion. The authors should, either wait for publication of ref. 36, or send a copy of this paper, if it is actually accepted, to the Referees.
Response 3. We would like to clarify the situation with work [36] ([32] in revised version of the manuscript). This work is a translation of an article that has already been published in Russian. The publisher has already prepared an English version of the article for printing. The reference mentioned in the manuscript is correct. As the English version of the article has not yet been included in the Scopus and Web-Of-Science databases, you can use a copy of the article sent to us by the publisher. A file with a copy of the English version of the article is attached.
There are also some less important criticisms, which have also to be answered.
Point 4. Although the images of the NPs after H2 expoure dispaly a good resolution (fig.4), the resoultion is much poorer in fig. 2 and 3. It shoud be necessary to show topogrtaphic and current images with the same resolutions than in Fig.4 for small and large GNP, without H2 exposure.
Response 4. Fig. 2 and Fig. 3 are parts of larger images. The remaining parts of large images are not informative. For this reason, their resolution is slightly worse than the resolution of Fig. 4. We considered it useless to provide images of a large surface area, only a small part of which is important.
Point 5. Tunneling conditions for each image have to be given (STM bias, STM current).
Response 5. We agree with the comment about the tunneling conditions for each image (STM bias, STM current). Appropriate additions have been made to the text of the article (see figure captions).
Point 6. The reference have to be checked : ref. 25, 27, 34 and 35 are the same : ref. 26 and 28 are the same.
Response 6. The list of references was revised. Appropriate corrections have been added to the manuscript.
Point 7. Finally, the English writing is not very good and it should be improved.
Response 7. Spelling and grammar were improved in accordance with your recommendations.

Reviewer 2 Report
This is an important contribution and should be published. To indirectly measure H-adsorption via current measurements is not a new concept (see ref 37), but nonetheless the quality of the data in the manuscript is high. The author major claim is that between 5 and 10 nm is the threshold where H-adsorption is inhibited.
I have a couple of questions that (if possible) the authors should address in the revision:
Fig 2c (IVs curves, average if current-potential characteristics of Au vs its support, HOPG). Here I believe all that the authors are looking is a difference in contact geometry. Why would the Au particle that sits on HOPG look more conductive than HOPG itself? The 2 elements are in series…
I suspect that the STM tip to Au contact is very conductive, therefore the bottle-neck to charge transfer become the Au/HOPG contact. This contact will be large, hence its low resistance (and higher currents). Or curvature is the key?
To explain this as curvature or contact size, I would suggest the authors to show IVs taken on HOPG planes versus IVs taken at the edges between planes (where curvature is very large)
Fig4c: no changes to the particle conduction upon exposure to H. How do the authors confirm thatH does not adsorb? The lack of changes in conductance could simply mean that a larger junction is less sensitive to the presence or absence of defects near the Fermi level. This point should be addressed in the revised manuscript.
Author Response
Many thanks for your valuable comments (especially, Com.#3) that helped us to improve the quality of the manuscript.
Point 1. Fig 2c (IVs curves, average if current-potential characteristics of Au vs its support, HOPG). Here I believe all that the authors are looking is a difference in contact geometry. Why would the Au particle that sits on HOPG look more conductive than HOPG itself? The 2 elements are in series…
Response 1. At equal bias voltages and widths of the tunneling barrier (vacuum gap, the distance between the STM tip and the sample surface), the conductivity of the tunneling contact is determined by the local density of the electron states of the tip and the sample. The tip is stable during the experiments and its density of electron states does not change. So, the difference in the conductivity of a tunneling nanocontact in different areas of the sample is determined by the difference in the local electron density of the sample surface. Note, that the current tube of the tunneling nanocontact – the sample surface area of effective electron tunneling – is only a few angstroms in the diameter. So, it is indeed the local electron density that we detect by measuring the volt-ampere characteristics of STM nanocontact. Graphite is a semimetal, that is, it has a relatively low density of electron states near the Fermi level, and this is the key aspect in comparison with metals. Since we operate in the mode of relatively small bias voltages Vbias ≤ 2 V, the main contribution is made by the electron states located near the Fermi level. This explains the fact that the conductivity of the tip-graphite tunneling contact is slightly lower than the conductivity of the tip-gold tunneling contact. So, indeed Au nanoparticles supported on HOPG look more conductive than HOPG itself. Moreover, this is typical not only for gold nanoparticles, but also for nanoparticles of other metals. The geometry of a tunneling nanocontact can influence its conductivity, but in this case this effect is negligible.
Point 2. I suspect that the STM tip to Au contact is very conductive, therefore the bottle-neck to charge transfer become the Au/HOPG contact. This contact will be large, hence its low resistance (and higher currents). Or curvature is the key? To explain this as curvature or contact size, I would suggest the authors to show IVs taken on HOPG planes versus IVs taken at the edges between planes (where curvature is very large)
Response 2. We agree that there is an ohmic contact between the graphite support and the gold nanoparticles, but its conductivity is approximately 7-9 orders greater than the conductivity of the tunneling nanocontact between the STM tip and the gold nanoparticle. So, actually it is the tip-nanoparticle tunneling nanocontact that acts as the bottle-neck for charge transfer. As mentioned above, the conductivity of a tunneling nanocontact is determined by the local properties of the sample surface and the effect of curvature in this case is not significant.
Point 3. Fig4c: no changes to the particle conduction upon exposure to H. How do the authors confirm that H does not adsorb? The lack of changes in conductance could simply mean that a larger junction is less sensitive to the presence or absence of defects near the Fermi level. This point should be addressed in the revised manuscript.
Response 3. The method of scanning tunneling spectroscopy is sensitive to changes in the local electron density of the sample. Hydrogen adsorption significantly influences the electron density near the Fermi level, which determines the conductivity of the tunneling nanocontact under the conditions of our experiment. It was shown in previous report [Dokhlikova N.V., Kolchenko N.N., Grishin M.V., Shub B.R., Electron delocalization in heterogeneous AunHm systems // Nanotechnologies in Russia, 11 (2016) 7] that hydrogen adsorption leads to the very local change in the conductivity of a tunneling nanocontact. Indeed it is due to the property of locality, that we can detect hydrogen atoms adsorbed on the surface of nanoparticles. Nevertheless, we really cannot exclude the adsorption of a countable number of hydrogen atoms on the surface of large nanoparticles, e.g. on any defects. But this quantity will be significantly less in comparison to the case of small nanoparticles. Appropriate clarification has been added to the manuscript (lines 167-169 in revised manuscript).
Round 2
Reviewer 1 Report
The new version of the article proposed by Gatin et al is greatly improved compared to the previous version.
The main objections I had made have been answered satisfactorily and several paragraphs have been added to the document, which makes the discussion much more precise, sensible and convincing. In addition, English writing has been improved satisfactorily.
Therefore, I recommend the publication of the new version of this article as is.